# Resveratrol in Treating Diabetes and Its Cardiovascular Complications: A Review of Its Mechanisms of Action

**DOI:** 10.3390/antiox11061085

**Published:** 2022-05-30

**Authors:** Meiming Su, Wenqi Zhao, Suowen Xu, Jianping Weng

**Affiliations:** Department of Endocrinology, Institute of Endocrine and Metabolic Diseases, The First Affiliated Hospital of USTC, Division of Life Sciences and Medicine, Clinical Research Hospital of Chinese Academy of Sciences (Hefei), University of Science and Technology of China, Hefei 230001, China; sumeiming@mail.ustc.edu.cn (M.S.); zwq353063945@mail.ustc.edu.cn (W.Z.)

**Keywords:** resveratrol, diabetes mellitus, cardiovascular complications, insulin resistance, metabolism, anti-inflammation, anti-oxidative stress

## Abstract

Diabetes mellitus (DM) is one of the most prevalent chronic diseases worldwide. High morbidity and mortality caused by DM are closely linked to its complications in multiple organs/tissues, including cardiovascular complications, diabetic nephropathy, and diabetic neuropathy. Resveratrol is a plant-derived polyphenolic compound with pleiotropic protective effects, ranging from antioxidant and anti-inflammatory to hypoglycemic effects. Recent studies strongly suggest that the consumption of resveratrol offers protection against diabetes and its cardiovascular complications. The protective effects of resveratrol involve the regulation of multiple signaling pathways, including inhibition of oxidative stress and inflammation, enhancement of insulin sensitivity, induction of autophagy, regulation of lipid metabolism, promotion of GLUT4 expression, and translocation, and activation of SIRT1/AMPK signaling axis. The cardiovascular protective effects of resveratrol have been recently reviewed in the literature, but the role of resveratrol in preventing diabetes mellitus and its cardiovascular complications has not been systematically reviewed. Therefore, in this review, we summarize the pharmacological effects and mechanisms of action of resveratrol based on in vitro and in vivo studies, highlighting the therapeutic potential of resveratrol in the prevention and treatment of diabetes and its cardiovascular complications.

## 1. Introduction

Diabetes mellitus (DM), a prevalent metabolic disease, is characterized by β cell dysfunction, insulin secretion disorder, and hyperglycemia. DM is currently classified as type 1 diabetes mellitus (T1DM), type 2 diabetes mellitus (T2DM), gestational diabetes mellitus, and special type DM [1,2]. On December 6, 2021, the International Diabetes Federation (IDF) announced that diabetes has become one of the fastest-growing global health emergencies in the 21st century and has reached an alarming level as a major health problem. Statistics data show that in 2021, the number of adults with diabetes has reached 537 million globally, and about 10.5% of the world’s adults are affected by diabetes. Therefore, current data show that the prevalence of diabetes is still on the rise worldwide, and the health burden brought by diabetes is still a major challenge to individuals, families, and society [3]. Patients with diabetes usually not only present with hyperglycemia, but also many complications with high incidence and mortality, including diabetic nephropathy, diabetic eye complications, diabetic foot diabetes, cardiovascular complications, and diabetic neuropathy [4]. Among them, cardiovascular complications of diabetes are one of the common complications of diabetes, and one of the main causes of death in diabetes patients [5]. The treatment of diabetes center on reducing patient blood glucose level, and the commonly-used treatment means include lifestyle modification and anti-diabetic medications. At present, the mainstream drug therapy in the clinics includes insulin secretagogues agents, metformin, sodium glucose transporter 2 (SGLT-2) inhibitors, GLP1 receptor agonists, and α -glycosidase inhibitors [6]. These drugs generally have certain side effects, including hypoglycemia, gastrointestinal problems, or urinary infections. Considering the increasing number of diabetes patients worldwide, there is an urgent need to discover safe and effective complementary drugs that can display anti-hyperglycemic effects and potentially protect against diabetes complications.

Natural products have long been deemed as an eminent source of anti-diabetic drugs. As a representative natural product with multiple metabolic benefits, resveratrol (3,4′,5-trihydroxystilbene) is a polyphenol anti-toxin found in many plants, such as peanuts, berries, and grapes [7]. Resveratrol has been proved to be a strong antioxidant, which can prevent a wide range of diseases, such as cancer, diabetes, and cardiovascular diseases in various animal models [8,9,10]. Many studies have shown that resveratrol has certain protective effects against diabetes and its cardiovascular complications based on purported antioxidant, anti-inflammatory, hypoglycemic, activation of the SIRT1-AMPK signaling pathway, induction of autophagy, and other effects or molecular mechanisms to alleviate diabetes and cardiovascular complications [11,12,13]. This review aims to provide a comprehensive and up-to-date synthesis of the therapeutic effects of resveratrol in DM and its cardiovascular complications based on in vitro to in vivo studies. We also elucidate the potential molecular mechanisms of resveratrol, aiming to provide an overall understanding of resveratrol and provide a reference base for resveratrol in the treatment of DM and its cardiovascular complications.

## 2. Resveratrol and Diabetes

Studies of resveratrol in diabetes are mostly in vitro studies and animal model experiments, and it can be clearly seen that resveratrol has multiple protective effects against diabetes. The protective effects of resveratrol on diabetes in various animal models are summarized in Table 1 [14,15,16,17,18,19,20,21,22,23,24,25,26,27,28,29,30,31,32]. For example, resveratrol enhances glucose uptake and metabolism, enhances pancreatic beta-cell protection, and improves insulin resistance. There are few studies on resveratrol in clinical trials [33,34,35]. Although resveratrol has been shown to have some benefits for diabetics, there have been some conflicting results. The possible reason lies in the dose and absorption of resveratrol. The specific molecular mechanism of resveratrol on blood glucose needs to be further explored. In the following sections, we will elucidate the anti-diabetic effects and molecular targets of resveratrol.

### 2.1. The Activation of SIRT1

Sirtuin1 (SIRT1) is a class III histone deacetylase of the Sirtuin family, which plays an important role in regulating metabolism and cardiovascular diseases and is a potential target for the treatment of various diseases [36]. Resveratrol can activate and upregulate NAD+ dependent SIRT1 [37], thereby improving or delaying the development of diabetes, cardiovascular disease, cancer, and other diseases [38,39,40]. SIRT1 plays a crucial role in the regulation of many downstream key proteins which impact glucose metabolism, including forkhead transcription factor O1 (FOXO1), endothelial nitric oxide synthase (eNOS), peroxisome-proliferating-activated receptor (PPAR)-α/γ and co-activator (PGC)-1α [41,42]. Kim et al. found that the acetylation of FOXOs proteins was reversible and SIRT1 could bind to FOXO1, FOXO3a, and FOXO4 protein, respectively, specifically removing the acetyl group of FOXOs, thus up-regulating the DNA binding ability of FOXOs protein to specific target gene promoters and increasing its transcriptional activity [43]. The occurrence of diabetes can lead to a decrease in SIRT1 activity and its expression, and then the enhanced acetylation of FOXO1 will lead to a significant increase in blood glucose in vivo after FOXO1 activation, which may ultimately aggravate insulin resistance [44]. In line with this, knockdown of FOXO1 in mouse adipose tissue improved insulin resistance [45,46]. Therefore, it can be speculated that resveratrol can inhibit FOXO1 expression through SIRT1, thereby improving insulin resistance and restoring normal blood glucose levels [47]. Interestingly, studies have found that resveratrol regulates PI3K/Akt pathway by activating SIRT1, inhibits FOXO1 activity and expression by blocking its dephosphorylation, and ultimately improves gluconeogenesis [48]. It has been reported that resveratrol can competitively inhibit phosphodiesterase 4 (PDE4), resulting in increased cAMP content [49]. Protein kinase A (PKA) activates the cAMP response element-binding protein (CREB), which subsequently activates the expression of PGC-1, thereby reducing oxidative stress [50,51].

### 2.2. The Activation of AMPK

AMPK (Adenosine 5-monophosphate (AMP)-activated protein kinase) is a fuel sensor in the regulation of energy metabolism and a star molecule in recent studies on diabetes and other metabolism-related diseases [52]. Resveratrol can activate AMPK and act on FOXO1 and PGC-1α through AMPK [53]. It has been shown in the literature that different doses of resveratrol have different effects on the interaction between AMPK and SIRT1. Specifically, in a dose of resveratrol less than 25 µM, AMPK was completely dependent on SIRT1, while when the dose of resveratrol was increased to 50 µM, AMPK seemed to function independently of SIRT1 [54]. Canto et al. demonstrated that AMPK increased NAD^+^ by stimulating its synthesis, which subsequently activated SIRT1 and many downstream efficient factors [55]. Likewise, another study showed that AMPK could promote the expression of NAMPT, an enzyme responsible for NAD+ biosynthesis, and NAMPT could further indirectly activate SIRT1 to play its role [56]. At the same time, resveratrol was found to activate SIRT1, which further activates AMPK’s upstream kinase threonine protein kinase liver kinase B1 (LKB1) [57]. These experimental data further confirmed the interaction between AMPK and SIRT1. Resveratrol has also been shown to activate AMPK and SIRT1 by inhibiting phosphodiesterase (PDE) activity. This pathway is accomplished through the increase in cAMP, which promotes signaling cascades [58]. It was verified in PDE3B knockout mice and PDE4B knockout mice, respectively, that knockout mice had higher insulin secretion levels and lower fat content compared with wild-type mice [59,60]. In conclusion, it is plausible that activation of the PDE-cAMP-AMPK-SIRT1 signaling pathway may be a new approach for resveratrol treatment of diabetes.

### 2.3. Anti-Oxidant Effects

Oxidative stress refers to the imbalance between the oxidative system and the antioxidant system of cells. The accumulation of a burst of ROS in cells that cannot be cleared leads to oxidative stress response in the body, resulting in tissue and cell damage [61]. At present, it is known that oxidative stress is one of the major contributors to diabetes and its complications as well as the pathogenesis of cardiovascular diseases [62,63]. Resveratrol is known to exert anti-oxidant effects in a variety of mechanisms: (1) scavenge free radicals; (2) reduce the generation of ROS; (3) activate the production of endogenous antioxidant enzymes; (4) promote the expression of antioxidant molecules through a variety of signaling pathways; (5) induce autophagy [64]. In the case of hyperglycemia and dyslipidemia, NADPH oxidase (NOXs) is activated and eNOS is inhibited in the body, and metabolic abnormalities increase the accumulation of advanced glycation end products (AGEs) and the generation of lipid toxicity [65]. It further increases the oxidative stress response in the body, aggravates insulin resistance in diabetic patients, destroys mitochondrial function, and leads to a series of malignant events such as β cell dysfunction [66]. Oxidative stress produces a large number of ROS, which can inactivate various protective factors, including SIRTs, AMPK, and other signaling pathways as well as FOXO, PGC-1α, and other protein factors. Nuclear factor E2-related factor 2 (NRF2) is a significant regulator of the body’s antioxidant defense system, which can enhance cell resistance to oxidative stress [67,68,69]. Bagul et al. reported that resveratrol could increase the protein level of NRF2 and downstream gene expression in the liver of high-fructose-fed rats, thereby mediating the antioxidant effects of resveratrol [70]. Therefore, the antioxidant effects of resveratrol, which are shared among polyphenols, underlie the metabolic benefits of resveratrol.

### 2.4. Improvement of Insulin Resistance

It has been shown that resveratrol treatment can improve insulin sensitivity in mice [20]. Insulin resistance was monitored using a homeostasis model assessment, and the results further confirmed that resveratrol can improve insulin resistance through activation of the AMPK signaling pathway [20]. Mounting evidence has shown that activation of SIRT1 can improve insulin sensitivity in the liver and adipose tissues and improve insulin resistance [71]. Luo and colleagues assessed the role of resveratrol insulin resistance induced with ethanol gavage in Sprague-Dawley rats. After 22 weeks of treatment, both SIRT1 and NAD^+^/NADH were down-regulated in the ethanol treatment group, while resveratrol ameliorated these down-regulation and improved insulin resistance [72]. These results suggest that resveratrol may regulate NAD^+^/NADH to stimulate SIRT1 for the treatment of diseases. Resveratrol also protects mitochondrial function and improves insulin resistance in obese mice by increasing PGC-1α activity [21]. Mitochondrial function is very important for insulin resistance in diabetes mellitus, and activation of SIRT1 and downstream target factors may be beneficial to the protection of mitochondrial function [18]. Asadi et al. found that FOXO3a content in adipose tissues was closely associated with SOD activity. The results showed that the content of FOXO3a decreased after resveratrol treatment, which further increased SOD activity and ultimately improved insulin resistance [47]. Resveratrol also plays a role in fructose-fed rats and improves insulin sensitivity [67]. It has been suggested that the mechanism of resveratrol improving insulin resistance may also be related to the inhibition of protein tyrosine phosphatase (PTP1B) transcription [73]. As a negative regulator, PTP1B has been confirmed to play an important role in the insulin signaling pathway [74]. To demonstrate the effect of resveratrol in diabetes in the clinics, Liu et al. conducted a comprehensive literature search and analysis to evaluate the effects of resveratrol on lowering blood glucose and improving insulin sensitivity in 11 studies involving 388 diabetic and non-diabetic patients. The results showed that resveratrol could significantly reduce patients’ blood glucose and improved insulin sensitivity, without obvious adverse reactions in non-diabetic patients [75].

### 2.5. The Enhancement of Glucose Uptake and Metabolism

Diabetic patients have obvious characteristics of glucose metabolism disorder, but normal glucose metabolism is crucial to maintaining a normal physiological state of the body. Skeletal muscle cells are the main contributors to maintaining the balance of glucose metabolism in the body [76,77]. There is extensive literature supporting that glucose transporter 4 (GLUT4) plays a key role in the uptake of glucose in skeletal muscle cells. The increase in glucose uptake induced by resveratrol also mainly depends on stimulating the expression and translocation of GLUT4, and the translocation of GLUT4 is mainly caused by the translocation of GLUT4 in fat and muscle cells from intracellular to the cell membrane [23,78,79]. Studies have shown that resveratrol-fed db/db mice (a classic insulin-resistant mouse model of type 2 DM) significantly increased glucose uptake increasing the level of GLUT4 [80]. It has been reported that resveratrol combined with insulin can improve the translocation of GLUT4 and thus glucose uptake in diabetic rats more than resveratrol or insulin treatment alone [81]. Resveratrol also increases the phosphorylation of AMPK by activating or binding to an estrogen receptor (ER), further increasing GLUT4 expression and translocation, thus affecting glucose uptake by skeletal muscle cells [82,83]. In addition, resveratrol improves glucose uptake in skeletal muscle by affecting protein kinase C-θ (PKC-θ) or through the PI3K-Akt pathway [23,27].

### 2.6. Regulatory Mechanism for Preventing β-Cell Dysfunction

Another significant function of resveratrol is its protective effects on islet β cells. It has been demonstrated that increased SIRT1 expression can effectively protect human islet β cells. In pancreatic β cells, SIRT1 was positively correlated with insulin secretion, whereas inhibition of SIRT1 expression reduced insulin secretion. Activation of SIRT1 inhibits the expression of intracellular uncoupling protein-1 (UCP-1), further leading to increased insulin secretion and improved insulin resistance [84]. Chen et al. investigated the effects of resveratrol on islet β cells by measuring ion channels and membrane potential. The results showed that resveratrol promoted insulin secretion by inhibiting the ATP-mediated K^+^ channel (KATP) and volt-mediated K^+^ channel (KV) in the cell membrane [85]. Bordone et al. found that SIRT1 enhances insulin secretion by decreasing the level of UCP2 in mouse islet β cells [84]. Much of the literature has proved that resveratrol potentiates insulin secretion, but some literature yields conflicting results, i.e., resveratrol inhibits insulin secretion. Szkudelski et al. studied the effect of resveratrol on insulin secretion in pancreatic tissues and found that resveratrol could inhibit pancreatic insulin secretion [86]. Another study also showed that resveratrol at different concentrations can affect insulin release in rat pancreas in vitro, and the inhibition of insulin secretion induced by resveratrol may be partially eliminated by using protein kinase C or acetylcholine [87]. Studies have shown that resveratrol’s inhibition of insulin secretion is beneficial in patients with diabetes. The molecular mechanisms of resveratrol may be through lowering ATP levels or modulating metabolic disorders to protect pancreatic function in patients with diabetes [86,88].

### 2.7. The Induction of Autophagy

Autophagy is a process of engulfing one’s own cytoplasmic proteins or organelles and making them encapsulated into vesicles, fusing with lysosomes to form autophagic lysosomes. The consequence of autophagy is to degrade the contents of their wrapping, and through this process realize the metabolic needs of the cells and the renewal of certain organelles [89]. Defective autophagy often occurs in diabetic conditions. In this scenario, resveratrol mitigated oxidative damage through AMPK-mediated inhibition of Mechanistic Target of Rapamycin Kinase (mTOR) and combated oxidative stress by inducing autophagy through AMPK-mediated activation of transcription factor EB (TFEB). This promotes the formation and fusion of autophagosomes and lysosomes into autophagic lysosomes [90,91]. In addition, resveratrol also inhibits the activity of NLRP3 inflammasome and upregulates the expression of the AMPK-SIRT1 signaling pathway to reduce key proteins of the MAPK signaling pathway and ultimately induce autophagy [64,92]. Therefore, the autophagy induction mechanism of resveratrol underlies resveratrol-mediated protective effects under metabolic stress conditions.

### 2.8. The Regulation of Lipid Metabolism

Diabetes is associated with deregulated glucolipid metabolism. In this regard, resveratrol plays an important role in lipid metabolism, by inhibiting the expression of genes associated with de novo lipogenesis and lipid deposition. Many studies have found that resveratrol may improve lipid accumulation in the liver by altering the expression of genes associated with fatty acid synthesis and transport [93,94,95]. Furthermore, resveratrol can also increase mitochondrial oxidation and reduce lipid accumulation in skeletal muscles [79]. Resveratrol also activates PGC-1α through the AMPK-SIRT1 signaling pathway, further activating PPAR and thus playing a role in lipid metabolism [96]. In addition to the liver and skeletal muscle, resveratrol also improves the metabolism of white adipose tissue in rhesus monkeys [97].

## 3. Resveratrol and Cardiovascular Complications of Diabetes

Diabetes is characterized by hyperglycemia and insulin resistance in metabolic tissues. The complications of diabetes are even more deleterious than hyperglycemia per se. Patients with diabetes are more likely to suffer from cardiovascular complications and eventual death [98,99]. In this regard, resveratrol has a good therapeutic effect on cardiovascular complications of diabetes in animal experiments. The protective effect of resveratrol on cardiovascular complications of diabetes in various animal models is shown in Table 2 [100,101,102,103,104,105,106,107,108,109,110,111,112,113,114]. Resveratrol plays an active role in diabetes and cardiovascular complications of diabetes through various signaling pathways (Figure 1). The antioxidant and anti-inflammatory effects of resveratrol play an important role in protecting endothelial cells, smooth muscle cells, and other important cell types in blood vessels. In addition, resveratrol has been shown to improve mitochondrial function. In a handful of clinical trials, resveratrol has been shown to inhibit the expression or secretion of inflammatory factors. In the following section, we provide a mechanistic review of resveratrol in protecting against cardiovascular complications associated with diabetes.

### 3.1. Activation of SIRT1

In one study, the authors treated type 1 and type 2 diabetic rats with resveratrol. The results showed that in rat hearts, resveratrol was able to inhibit alterations in SIRTs induced by streptozotocin (STZ) injection concurrent with high-fructose diet feeding. SIRT1 activation mediates the effect of resveratrol in the treatment of type 1 and type 2 diabetes [115]. Given the important role of FOXOs in cardiac endothelial function in diabetic cardiomyopathy (DCM) [116]. SIRT1 deacetylates FOXO1 and stimulates antioxidant production to inhibit oxidative stress response [117,118]. In addition, SIRT1 can also promote the expression of various antioxidant enzymes to combat oxidative stress responses in diabetic cardiomyopathy, such as manganese superoxide dismutase (MnSOD), through deacetylation of FOXO3a [119,120]. Another study has shown that resveratrol can activate SIRT1 and PI3K/Akt pathways, thus inhibiting the accumulation of FOXO3 and improving the cardiac function of rats [121]. In addition to FOXO3a, literature has shown that FOXO4 also plays a role in diabetic vascular complications and further inhibits nuclear factor kappa-B (NF-κB)-mediated pro-inflammatory responses through binding with SIRT1 [122]. In addition, resveratrol also promotes the activation of sarcoplasmic calcium ATPase, thereby protecting the myocardial function of diabetic cardiomyopathy [105]. Resveratrol has also been reported to improve mitochondrial function and ultimately restore myocardial function in diabetic rats through activation of SIRT1 and downstream factor PGC-1α [107]. Literature has shown that PPARα activity is compromised under hyperglycemia conditions, and SIRT1 can enhance antioxidant capacity by activating PPARα activity and improve diabetic cardiovascular complications [123]. In addition to affecting PPARα activity, Cheang et al. elegantly demonstrated that resveratrol also ameliorates endothelial dysfunction in diabetic mice by upregulation of the SIRT1/PPARδ pathway [124].

SIRT1 also plays an important role in protecting against cardiomyocyte apoptosis. It is well-established that ROS generation was increased in diabetic cardiomyopathy mainly by increased activity of NOXs, and the ROS generation further induces myocardial cell apoptosis [125]. In one study, diabetic rats developed myocardial hypertrophy and increased oxidative stress after 56 days of fructose-rich diet feeding. Resveratrol treatment further inhibits cardiac adverse symptoms by inhibiting NOXs by activating SIRT1, which leads to the deacetylation of NF-κB p65 subunit and histone 3 (H3) [125]. In line with this evidence, another study also showed that resveratrol improved oxidative stress by inhibiting NOXs and phosphorylating AMPK, further enhancing the protective effect on the heart [126]. 

In addition, previous studies have shown that SIRT1 can deacetylate endothelial nitric oxide synthase (eNOS), thus increasing NO availability and endothelium-dependent vasodilation [127]. As impaired NO production and vasodilation are associated with diabetes-accelerated endothelial dysfunction, resveratrol is presumed to exert protective effects via upregulating SIRT1-mediated eNOS-dependent vasodilatory effects [128,129].

### 3.2. Activation of AMPK

Numerous studies have shown that resveratrol can activate AMPK, and AMPK participates in the antioxidant stress response by interacting with SIRT1 alone or jointly regulating numerous downstream effector molecules, thus enhancing the protective effect of the heart [42,126]. AMPK alone or in conjunction with SIRT1 stimulates downstream PGC-1α activation and alleviates cardiac endothelial function damage [130]. AMPK activates FOXOs through phosphorylation or co-action with SIRT1, triggering the expression of antioxidant enzymes to restore cardiac function [55]. Resveratrol can indirectly enhance the activity of PPARα by activating AMPK, SIRT1, and PGC-1α, thereby inhibiting NF-κB and attenuating oxidative stress and inflammation [131,132,133,134]. Collectively, resveratrol can activate multiple pathways fostering the cooperation between AMPK and SIRT1, which plays a beneficial role in improving diabetic cardiovascular disease. For example, AMPK-SIRT1-PGC-1α, AMPK-SIRT1-FOXOs, and AMPK-SIRT1-PPARα pathways. AMPK activation has also been shown to play an important role in regulating glucose uptake in cardiac glucose metabolism [135,136]. In addition, for diabetic cardiomyopathy, AMPK also plays a critical role in cardiac autophagy [137].

### 3.3. Anti-Oxidant Effects

Resveratrol can reduce the oxidative stress response to treat cardiovascular complications of DM through multiple mechanisms. Endothelial cells, smooth muscle cells, and macrophages play a critical role in maintaining the homeostasis of the vascular environment [138]. eNOS participates in the protection of atherosclerosis in these cells by producing NO. It has been reported that resveratrol can maintain endothelial homeostasis through the increase in eNOS-derived NO bioavailability, via activation of SIRT1 or AMPK in endothelial cells [139,140,141,142]. It has been reported that resveratrol is an activator of KLF2 and KLF4, which are well-known transcription factors regulating gene expression of eNOS, suggesting that resveratrol may increase the expression of eNOS in a KLF2/4-dependent manner [143,144,145,146]. It has also been reported that resveratrol can promote the production of NO by vascular smooth muscle cells, thus combating atherosclerosis and other cardiovascular diseases [147]. In diabetic cardiomyopathy, resveratrol inhibits oxidative stress response by reducing the activity of extracellular regulated protein kinase (ERK) [148]. In a mouse model of type 1 diabetes, resveratrol can up-regulate NRF2, a transcription factor that reduces oxidative stress, to afford myocardial protection [149]. As a polyphenol compound, antioxidant effects mediate a major portion of mechanisms explaining the protective effects of resveratrol against cardiovascular complications of diabetes.

### 3.4. Anti-Inflammatory Effects

Resveratrol protects against cardiovascular complications of diabetes by reducing inflammation. It has been reported that resveratrol can inhibit the activity of the NF-κB signaling pathway, thus alleviating vascular diseases associated with diabetes [21]. It is well known that NF-κB can cause the activation of many pro-inflammatory factors, such as interleukin 1 beta (IL-1β) and tumor necrosis factor-alpha (TNF-α). Resveratrol can reduce the activation of NF-κB induced inflammatory factors, thus playing an anti-inflammatory role in the cardiovascular complications of diabetes [150]. It has been reported that resveratrol can activate the expression of KLF2 and reduce the expression of inflammatory factors such as IL-1β, intercellular adhesion molecule 1 (ICAM-1), vascular cell adhesion molecule 1 (VCAM-1), and TNF-α in endothelial cells [151,152,153]. Both endothelial cells and macrophages play an important role in maintaining vascular homeostasis. Resveratrol has also been shown to have anti-inflammatory effects by altering the activity of macrophages. Studies have shown that resveratrol protects cardiomyocytes and repairs heart damage by inhibiting mRNA and protein expression of NALP3 (NLRP3) inflammasome and caspase1, as well as down-regulating a variety of inflammatory cytokines, such as IL-1β and interleukin-18 (IL-18) [154]. Wu et al. investigated whether resveratrol can improve diabetic cardiomyopathy by regulating high mobility group box 1 (HMGB-1) in vivo. The results showed that resveratrol decreased HMGB-1 content in diabetic mice, and resveratrol significantly prevented the expression of cardiac fibrosis and inflammation in diabetic mice. HMGB-1 signaling pathway has been reported to be associated with inflammatory response and myocardial fibrosis, therefore, resveratrol may improve diabetic cardiomyopathy by downregulating the HMGB-1 signaling pathway [9]. The protective effect of resveratrol on cardiovascular complications of diabetes has also been seen in Rhesus monkeys, as resveratrol can reduce the expression of inflammatory factors in the inner wall of blood vessels, such as VCAM-1, ICAM-1, and MCP-1 [155]. In a clinical study, subjects were given either resveratrol-containing grape skin extract or a trans-resveratrol. After the completion of the trial, plasma resveratrol content was negatively correlated with the secretion of pro-inflammatory factors, such as VCAM-1, ICAM-1, and IL-8 [156].

### 3.5. Improvement of Mitochondrial Function

Mitochondrial dysfunction is one of the major causes of many cardiovascular diseases, including diabetic cardiomyopathy. It has been reported that mitochondria function was impaired under hyperglycemia conditions [126]. As mentioned above, resveratrol can improve the mitochondrial function of rats by activating the SIRT1 pathway [107]. Lagouge et al. showed that resveratrol reduces PGC-1 α acetylation and increases its activity by activating SIRT1, thereby affecting mitochondrial oxidative phosphorylation and expression of mitochondria protection-related genes, and restoring mitochondrial function [21]. In addition to SIRT1 activation, studies have also shown that resveratrol can also impact SIRT3, another member of the SIRTs family specifically localized in the mitochondria, thus conferring a protective effect on mitochondrial function. In a study of the type 2 DM model, SIRT3 was activated after treatment with resveratrol, altering the activity of mitochondrial transcription factor (TFAM) and ultimately improving mitochondrial function in diabetic hearts. This evidence suggests that SIRT3 may mediate the protective effects of resveratrol in attenuating cardiovascular complications of diabetes [114]. In another study, Lekli et al. found that resveratrol alleviates cardiac injury through up-regulation of GLUT4 expression [157]. In addition, glucose metabolism enzymes, such as glycogen synthase kinase 3β and aldose reductase, are thought to be regulated by resveratrol [158,159]. Resveratrol also protects the heart by down-regulating both enzymes to regulate the openness of mitochondrial permeability conversion pore (mPTP) [158,159].

### 3.6. Regulation of Lipid Metabolism

Deregulated lipid metabolism plays an important role in the development of diabetic cardiovascular disease. In one study, Yagyu et al. constructed transgenic mice that can normally express human non-metastatic lipoprotein lipase (LpL) in mouse cardiomyocytes and studied the effect of LpL on cardiomyocytes. The results showed that the hearts of the LpL transgenic mice accumulated more fat and had impaired heart function, resulting in cardiomyopathy [160]. Lipid accumulation will aggravate the progression of diabetic cardiomyopathy, leading to myocardial fibrosis and ultimately myocardial necrosis [161]. Beaudoin et al. treated obese type 2 diabetic adipose rats (ZDF rats) with resveratrol and found that resveratrol can reduce the accumulation of lipid in the heart, slow down the degree of myocardial fibrosis and improve the progression of diabetic cardiomyopathy by inhibiting the sensitivity of enzymes related to lipid metabolism, such as palmitoyl-CoA (P-CoA) [110].

### 3.7. Induction of Autophagy

Recently, the biological mechanism of autophagy has attracted much attention in many diseases including diabetic cardiomyopathy. Wang et al. [118] found that resveratrol can improve cardiac dysfunction. The authors used resveratrol to treat STZ-induced diabetic cardiomyopathy in mice and found that resveratrol improves dysfunctional autophagy flux through the SIRT1-FOXO1-RAB7 pathway, which may provide a new approach for the treatment of diabetic cardiomyopathy [118]. Studies have also shown that autophagy activity in the heart is enhanced in type 1 diabetic cardiomyopathy and reversed in type 2 diabetic cardiomyopathy. Although there are significant differences in heart structure between type 1 diabetic mice and type 2 diabetic mice, such as lysosomes, resveratrol can simultaneously improve the diastolic function of type 1 and type 2 diabetic mice, enhance cardiac autophagy, and inhibit cardiac hypertrophy and fibrosis [162]. Hyperglycemia and free fatty acid such as palmitic acid are contributors to diabetes and its complications. Both hyperglycemia and palmitic acid have been reported to inhibit cardiac autophagy and accelerate the process of apoptosis. After treatment with resveratrol, cardiomyocytes restore autophagy and alleviated apoptosis by activating the interaction of the two signaling pathways. The two pathways are AMPK-mediated phosphorylation of mTOR/p70S6K1/4EBP1 and c-Jun *N*-terminal protein kinase 1 (JNK1) -mediated dissociation of the Beclin1-Bcl-2 complex. [137]. Autophagy has been known to be associated with the activation of AMPK and SIRT1 and the inhibition of mTOR signaling pathways [163,164,165]. Some studies have shown that excessive autophagy is not conducive to the protection of the heart, and even leads to the occurrence of cardiomyopathy and other diseases, suggesting that autophagy should be finely tuned to avoid unwanted side effects [166]. In conclusion, autophagy targeted therapy has potential application in preventing diabetic cardiomyopathy and resveratrol appears to be a promising candidate drug.

### 3.8. Other Molecular Mechanisms

On top of the reviewed mechanisms as mentioned above, there are other mechanisms that may be accountable for the protective effects of resveratrol. In one study, Ding et al. [167] discovered a new pathway (E2F3 pathway) essential for the protection of vascular endothelial cells. The researchers found that this pathway was inhibited in vascular endothelial cells exposed to high glucose. In contrast, activation of this pathway confers protective effects. The authors also showed that the vascular damage induced by high glucose is reduced by activation of the E2F3 pathway upon resveratrol treatment. This may provide a new direction and target for the treatment of diabetic cardiovascular disease [167]. Ashrafizadeh et al. found that resveratrol can mitigate inflammatory response and reduce fibrosis by inhibiting the expression of transforming growth factor β (TGF-β) [168]. TGF-β activation has been well established to promote the development of cardiovascular diseases, diabetes, and diabetes complications, as well as fibrotic disorders [169,170]. 

## 4. Concluding Remarks and Future Perspectives

Resveratrol is a representative natural product that displays pleiotropic protective effects against a variety of diseases, including cardiovascular complications, but the role of resveratrol in diabetes and the cardiovascular complications of diabetes currently has not been systematically reviewed, so this article synthesizes the therapeutic effects of resveratrol in diabetes mellitus and cardiovascular complications of diabetes mellitus and summarizes the specific mechanisms of direct or indirect molecular targets. According to the current literature, resveratrol exerts its effects by anti-oxidative stress, anti-inflammatory, the protecting of islet β cells, the improvement of mitochondrial function, the regulation of lipid metabolism, the production of autophagy, and so on. Resveratrol can improve diabetes and cardiovascular complications by regulating multiple signaling pathways, especially the SIRT1 and AMPK signaling pathways. Given the hypoglycemia and other side effects of existing drugs for treating diabetes and its cardiovascular complications, resveratrol may be used as an alternative or in combination with other standard anti-diabetic drugs. The disadvantage is that although resveratrol has been widely reported in animal experiments, only a few have been reported in clinical trials (Table 3). Moreover, we have to note that previous pre-clinical studies used varied doses of resveratrol in experimental animals, the varied doses of resveratrol in these studies may be due to different routes of administration, low oral bioavailability, and inconsistent targets. Therefore, the effective dose of resveratrol in vivo needs to be optimized in different models after considering the oral bioavailability of resveratrol. In addition, in cultured cells, the dose of resveratrol is supraphysiological, data from cultured cells need to be interpreted with caution. Therefore, further research is needed to clarify the precise mechanism of action of resveratrol to treat patients with diabetes at therapeutically relevant concentrations. Further structural modification of resveratrol and the development of sustained-release dosage forms are necessary. In conclusion, resveratrol represents a promising nutraceutical against diabetes and its cardiovascular complications. Large-scale randomized clinical trials are warranted to confirm the therapeutic potential of resveratrol in diabetic patients.

## Figures and Tables

**Figure 1 antioxidants-11-01085-f001:**
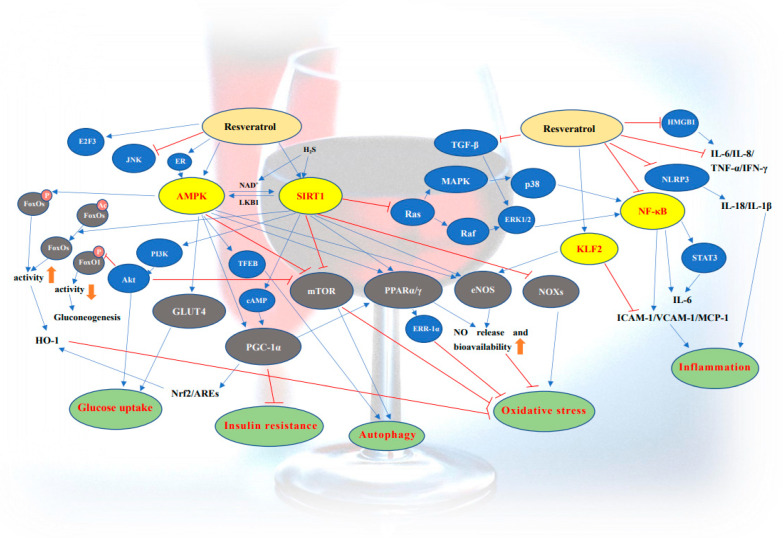
Role of resveratrol in diabetes mellitus and cardiovascular complications.

**Table 1 antioxidants-11-01085-t001:** Effects of resveratrol in diabetes.

Study Type	Model	Dose/Dosing Method/Period	Outcome	Proposed Mechanism	Ref.
In vivo	SD rats (STZ DM model)	RES 0.5 mg/kg, gavage for 8–14 days	↓Insulin resistance ↑Glucose uptake ↑Hepatic glycogen synthesis		[14]
In vivo	Wistar rats (STZ-NA model)	RES 5 mg/kg, oral for 30 days	↓Blood glucose ↓Plasma insulin and hemoglobin ↓AST, ALT, ALP		[15]
In vivo	*db/db* mice (T2DM model)	RES (0.3% mixed in chow) for 8 weeks	↑Mitochondrial oxidative stress and biogenesis ↓Blood glucose	RES improves oxidative stress and promotes mitochondrial biogenesis through normal Mn-SOD function and glycolipid metabolism.	[16]
In vivo	C57BL/6 mice (HFD)	RES 0.03 µg/µL minipump Intracerebroventricularly, 14 weeks	↓Hyperglycemia ↓Pyruvate-induced hyperglycemia	RES improves hypothalamic NF-κB inflammatory signal transduction by decreasing total and acetylated RelA/P65 protein content.	[17]
In vivo	*ob/ob* mice (T2DM model)	RES 5, 15, 50 mg/kg, oral for 4 weeks	↓Hyperglycemia ↓Insulin resistance ↓TG, TC, ADPN, FFA		[18]
In vivo	NOD mice (T1DM model)	RES 250 mg/kg oral or subcutaneously inject for 32 weeks	↓Expression of inflammatory genes ↓Expression of CCR6	RES blocks CCR6 and CD11b (+) F4/80(hi) macrophages migration from peripheral lymphoid organs to the pancreas.	[19]
In vivo	C57BL/6 mice (HFD)	RES (0.04% mixed in chow) for 6 months	↑Survival ↓Insulin sensitivity ↑Mitochondrial number	RES reduces IGF-I levels and increases AMPK and PGC-1α activity.	[20]
In vivo	C57BL/6 mice (HFD)	RES 400 mg/kg, oral for 16 weeks	↓Insulin resistance ↑Mitochondrial biogenesis ↑Oxidative phosphorylation	RES improves mitochondrial function and protects against metabolic disease by activating SIRT1 and PGC-1α.	[21]
In vivo	SD rats (HCF)	RES 1 mg/kg, oral for 15 days or 15 weeks	↑Glucose uptake ↑Membrane trafficking activity of GLUT4 ↑Phosphorylation of insulin receptor	ER is a key regulator in RES-stimulating insulin-dependent and -independent glucose uptake.	[22]
In vivo	Wistar rats (STZ/STZ-NA/ insulin-resistant diabetic model)	RES 3 or 10 mg/kg, oral for 90 min	↓Blood glucose ↓Insulin resistance ↑GLUT4 expression	RES promotes skeletal muscle glucose uptake through the PI3K-Akt signaling pathway.	[23]
In vivo	NOD mice (T1DM model)	RES 200 mg/kg, gavage for 28 days	↓Blood glucose ↓Inflammatory factors	RES improves renal function not only by its anti-inflammatory effect but also by improving the metabolic memory of hyperglycemia.	[24]
In vivo	SD rats (STZ model)	RES 5, 10 mg/kg, gavage for 1–7 months	↓Blood glucose ↑Weight	RES significantly inhibited the HG-induced decreases in glutamate uptake, GS activity, GLAST, and GS expression.	[25]
In vivo	Albino rats (Alloxan model)	RES 30 mg/kg, gavage for 30 days	↓Hyperglycemia		[26]
In vivo	ICR mice (HFD)	RES 50 mg/kg, oral for 10 days	↓HIF-1α ↓Inflammation in the adipose tissue ↓Insulin sensitivity	RES reduces cAMP accumulation by preserving PDE3B, thereby preventing PKA/HSL activation and lipolysis, and decreasing FFAs influx and DAG accumulation, thereby improving insulin signaling by inhibiting PKCθ translocation.	[27]
In vivo	Wistar rats (STZ model)	RES 5 mg/kg, oral for 8 weeks	↓Blood glucose ↑Antioxidant status	RES significantly improved the expression of TGF-β1, fibronectin, NF-κB/P65, Nrf2, Sirt1, and FoxO1 in the kidney.	[28]
In vivo	*db/db*, *db/m* mice (T2DM model)	RES 10 mg/kg, gavage for 12 weeks	↓Apoptosis of podocytes ↑Autophagy of podocytes	Resveratrol regulates autophagy and apoptosis of podocytes by inhibiting microRNA-383-5p.	[29]
In vivo	Wistar albino rats (STZ model)	RES 20 mg/kg, gavage for 8 weeks	↓Hyperglycemia ↓Serum MDA concentrations	Resveratrol inhibits oxidative stress and increases the potential of extra-hepatic tissues to absorb glucose.	[30]
In vivo	SD rats (HFS model)	RES 147.6 mg/kg, oral for 12 weeks	↓Dysregulated gluconeogenesis ↓Dysregulation of several metabolic genes		[31]
In vivo	ICR mice (STZ model)	RES 50 mg/kg, oral for 7 days	↓TXNIP/NLRP3 inflammasome activation ↓Cell apoptosis ↓ROS-associated mitochondrial fission	Resveratrol inhibits Drp1 activity to protect mitochondrial integrity and inhibits endoplasmic reticulum stress to prevent NLRP3 inflammasome activation.	[32]

ADPN: Adiponectin; AMPK: Adenosine 5-monophosphate (AMP)-activated protein kinase; ALP: Alkaline phosphatase; ALT: Alanine transaminase; AST: Aspartate transaminase; cAMP: Cyclic AMP; CCR6: Chemokine (C-C motif) ligand 6; DAG: Diacylglycerol; DM: Diabetes mellitus; Drp1: Dynamin-related protein 1; ER: Estrogen receptor; FFA: Free fatty acid; FoxO1: Forkhead transcription factor 1; GLAST: Glutamate transporters; GLUT4: Glucose transporter 4; GS: Glutamine synthetase; HCF: High cholesterol-fructose; HFS: High-fat and sucrose diet; HIF-1α: Hypoxia-inducible factor 1α; IGF-I: Insulin-like growth factor-1; MDA: Malondialdehyde; Mn-SOD: Manganese superoxide dismutase; NF-κB: Nuclear factor-kappaB; NLRP3: NOD-like receptor thermal protein domain associated protein 3; Nrf2: Nuclear factor E2-related factor; PDE3B: Phosphodiesterase 3B; PGC-1α: Peroxisome proliferator-activated receptor-gamma coactivator 1alpha; PI3K-Akt: phosphatidylinositol 3-kinase-Akt; PKCθ: Protein kinase Cθ; RES: Resveratrol; SIRT1: Sirtuin 1; STZ-NA: Streptozotocin and Nicotinamide; TC: Total cholesterol; TG: Triglycerides; TGF-β1: Transforming growth factor-beta1; TXNIP: Thioredoxin-interacting protein. ↑: Increase; ↓: Decrease.

**Table 2 antioxidants-11-01085-t002:** Effects of resveratrol in diabetic cardiovascular complications.

Study Type	Model	Dose/Dosing Method/Period	Outcome	Proposed Mechanism	Ref.
In vivo	SD rats (STZ DM model)	RES 2.5 mg/kg, oral 15 days	↑Phosphorylation of eNOS ↓Blood glucose	RES improves diabetic myocardial GLUT4 translocation and glucose uptake through the AMPK pathway and by regulating the status of Cav-1 and Cav-3.	[100]
In vivo	Wistar rats (STZ DM model)	RES 5 mg/kg, intraperitoneal inject 42 days	↑Contractile responses to noradrenaline ↑Relaxation response to Ach ↓Blood glucose		[101]
In vivo	C57BL/6 mice (HFD) and *db/db* mice (T2DM model)	RES 5, 30, 50 mg/kg, oral for 4 weeks	↓Plasma insulin levels ↓Hyperglycemia↓Fasting BP ↓Angiogenesis ↑Endothelial protection	RES protects diabetic wound healing through its SIRT1-dependent endothelial cell protection and pro-angiogenesis, involving inhibition of FOXO1 and de-inhibition of c-Myc expression.	[102]
In vivo	C57BL/6 mice (HFD) and *db/db* mice (T2DM model)	RES (0.3% mixed in chow) for 8 weeks	↓Blood glucose, FFA ↓ICAM-1, VCAM-1, MCP-1 ↓NF-κB activity	RES ameliorates diabetic vascular inflammation and macrophage infiltration by inhibiting the NF-κB pathway.	[103]
In vivo	SD rats (STZ model/HFD)	RES 10 mg/kg, gavage for 8 months	↓Insulin sensitivity ↓TG, TC, LDLc ↓ROS	UCP2 mediates RES to improve cardiac function, inhibit myocardial cell apoptosis, and participate in the improvement of mitochondrial function.	[104]
In vivo	CD1 mice (STZ T1DM model)	RES 100 mg/kg, oral for 3 months	↑SERCA2 promoter activity ↑SIRT1	RES enhances SERCA2a expression and improves cardiac function through activation of SIRT1.	[105]
In vivo	SD rats (STZ-NA model)	RES 5 mg/kg, oral for 4 months	↓Antioxidant enzymes activities ↓Oxidative markers	RES treatment may delay or attenuate the progression of diabetes-related cardiac complications by reducing oxidative stress.	[106]
In vivo	SD rats (HFD T2DM model)	RES 50 mg/kg, gavage for 16 weeks	↓Cardiac dysfunction and hypertrophy ↓SOD activity ↓ATP content	RES activates SIRT1 and increases PGC-1α deacetylation, thereby regulating mitochondrial function and alleviating cardiac injury in diabetic rats.	[107]
In vivo	mice (STZ T1DM model)	RES 25 mg/kg, intraperitoneal inject for 5 days	↓Apoptosis ↑Mitochondrial biogenesis	Activation of SIRT1 by RES ameliorates myocardial injury in DCM through PGC-1α -mediated mitochondrial regulation.	[108]
In vivo	SD rats (STZ T1DM model)	RES 80 mg/kg, intraperitoneal inject for 12 weeks	↑Glucose and lipid metabolism ↑Cardiac function ↓TNF-α, IL-6, IL-1β	Res alleviates cardiac dysfunction caused by diabetes through down-regulation of the AT1R-ERK/P38 MAPK signaling pathway.	[109]
In vivo	ZDF rats	RES 200 mg/kg, oral for 6 weeks	↑The apparent Km to palmitoyl-CoA ↓Mitochondrial reactive oxygen ↓Lipid accumulation	Resveratrol reduces liver fibrosis, p-COA respiratory sensitivity, active lipid accumulation, and mitochondrial reactive oxygen emission rates.	[110]
In vivo	Wistar albino rats (DHEA-induced PCOS model)	RES 20 mg/kg, oral for 28 days	↓Serum testosterone levels ↓Number of TUNEL (+) granulosa cells ↓Number of Graafian follicles ↓Body weights	Resveratrol activates SIRT1 and AMPK to induce antioxidant and anti-inflammatory systems of PCOS.	[111]
In vivo	ICR mice (HFD model)	RES 50 mg/kg, gavage for 7 days	↓Collagen deposition ↓HIF-1α accumulation ↓Fibrosis and inflammation	Resveratrol reduces HIF-1α accumulation by promoting proteasome degradation of HIF-1α by regulating AMPK/SIRT1.	[112]
In vivo	SD rats (STZ model)	RES 0.1, 1, 5, 10, 50 μg/kg, intravitreal inject or tail vein injects for 12 weeks	↑Insulin level ↓AGEs, LDL, Ox-LDL, caspase 3 activity ↓Damage of DR	Resveratrol reduces the inflammatory state and damage of DR through PON1.	[113]
In vivo	SD rats (STZ T1DM model)	RES 25 mg/kg, oral for 8 weeks	↓Cardiac cell size ↓Oxidative stress ↓Fibrosis	Resveratrol activates SIRT3, maintains mitochondrial function, and regulates the acetylation of TFAM.	[114]

Ach: Acetylcholine; AGEs: Advanced glycation end products; AMPK: Adenosine 5-monophosphate (AMP)-activated protein kinase; AT1R: AGTR1, Angiotensin II receptor type 1; ATP: Adenosine triphosphate; BP: Blood pressure; Cav-1: Caveolin 1; Cav-3: Caveolin 3; DHEA: Dehydroepiandrosterone; DM: Diabetes mellitus; DR: Diabetic retinopathy; eNOS: Endothelial nitric oxide synthase; FFA: Free fatty acid; FOXO1: Forkhead transcription factor 1; GLUT4: Glucose transporter 4; HIF-1α: Hypoxia inducible factor 1 subunit alpha; ICAM-1: Intercellular adhesion molecule 1; IL-1β: Interleukin 1 Beta; IL-6: Interleukin 6; LDL: Low density lipoprotein; LDLc: Low-density lipoprotein cholesterol; MAPK: Mitogen-activated protein kinase; MCP-1: CCL2, C-C motif chemokine ligand 2; NF-κB: Nuclear factor kappa B subunit 1; Ox-LDL: Oxidized low-density lipoprotein; p-COA: palmitoyl-CoA; PCOS: Polycystic ovary syndrome; PGC-1α: Peroxisome proliferator-activated receptor-gamma coactivator 1alpha; PON1: Paraoxonase 1; RES: Resveratrol; ROS: Reactive oxygen species; SERCA2: ATP2A2, ATPase sarcoplasmic/endoplasmic reticulum Ca2+ transporting 2; SIRT1: Sirtuin 1; SIRT3: Sirtuin 3; SOD: Superoxide dismutase; T1DM: Type 1 diabetes mellitus; T2DM: Type 2 diabetes mellitus; TC: Total cholesterol; TFAM: Recombinant transcription factor A, Mitochondrial; TG: Triglycerides; TNF-α: Tumor necrosis factor; UCP2: Uncoupling protein 2; VCAM-1: Vascular cell adhesion molecule 1. ↑: Increase; ↓: Decrease.

**Table 3 antioxidants-11-01085-t003:** Clinical trial of resveratrol in the treatment of diabetes mellitus and its cardiovascular complications.

Identifier No.	Type	Dose/Dosing Method/Period	Phase	Sex	Number Enrolled	Outcome Measures
NCT01038089	T2DM	RES (90 mg/d and 270 mg/d for 2 weeks)	Not Applicable	All	20	Brachial artery flow-mediated dilation Blood markers of inflammation, oxidative stress, insulin resistance
NCT01677611	T2DM	RES (3 g/d for 12 weeks)	Phase 1	Male	10	SIRT1 expression Skeletal muscle AMPK expression Skeletal muscle p-AMPK expression
NCT01881347	T2DM	RES (100 mg/d for 2 weeks and then 300 mg/d for 2 weeks)	Not Applicable	All	54	Change from baseline in Brachial artery flow-mediated dilation Change from Baseline in Fingertip peripheral arterial tonometry Change from Baseline in Carotid femoral pulse wave velocity Change from Baseline in Reactive hyperemia
NCT01638780	T2DM	RES (150 mg/kg/d for 30 days)	Not Applicable	Male	24	insulin sensitivity (overall, muscle- and liver-specific) muscle mitochondrial oxidative capacity intramyocellular lipid content
NCT04449198	T1DM	RES (500 mg, twice a day for 12 weeks)	Early Phase 1	All	24	Change in AUC for ET-1 + BQ-123 Skeletal Muscle Mitochondrial Function Change in Percentage FMD
NCT03436992	T1DM	RES (1500 mg for 3 months)	Not Applicable	All	198	Change in FMD
NCT03762096	T2DM+CAD	RES (1 g, twice a day for 6 weeks)	Not Applicable	All	40	Change in endothelial function Effects of resveratrol on caveolar function Effects of resveratrol on molecular signaling
NCT01354977	T2DM+ Insulin Resistance	RES (1000 mg, twice a day for 4 weeks)	Phase 2	All	20	Peripheral Insulin Sensitivity (RD) Measured by the Change in Glucose Rates of Disappearance with Resveratrol or Placebo at Baseline and at 4 weeks. EGP, With Resveratrol or Placebo at Baseline and at 4 weeks. Effects of Resveratrol on Skeletal Muscle Mitochondrial Numbers
NCT02244879	T2DM+ Inflammation+Insulin Resistance	RES (40 mg/d and 500 mg/d for 6 months)	Phase 3	All	192	CRP Metabolic and oxidative markers

Website: ClinicalTrials.gov (accessed on 19 May 2022). AUC: Area under the curve; CAD: Coronary artery disease; CRP: C reactive protein; EGP: Endogenous glucose production; ET-1: Endothelin 1; FMD: Flow-mediated dilation; p-AMPK: phosphorylated-AMPK-Thr172; RES: Resveratrol; SIRT1: Sirtuin 1; T1DM: Type 1 diabetes mellitus; T2DM: Type 2 diabetes mellitus.

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
