# Peer review of "Resveratrol in Treating Diabetes and Its Cardiovascular Complications: A Review of Its Mechanisms of Action"

_antioxidants, 2022, doi:10.3390/antiox11061085_

Round 1
Reviewer 1 Report
In the manuscript, “Resveratrol in treating diabetes and its cardiovascular complications: a review of its mechanisms of actions”, the authors preformed a review of literature. The review covers interesting and emerging topics of resveratrol on diabetes and cardiovascular disease. The authors did a good job in describing the articles they studied. They also outlined the possible mechanisms by which resveratrol has beneficial effects.
Minor corrections and editions throughout the text together with a nicely described conclusion could improve future research and publications on the desired topic, especially randomized clinical trials in humans.
Line 30 – Please change type I to type 1 and type II to type 2 which is the correct nomenclature used.
Line 55 – Please add that resveratrol has been proved to be a strong antioxidant …cardiovascular diseases in the animal model.
Line 72 – Please add references to denote the few studies on resveratrol in human clinical trials.
Thank you for allowing me to review the article. I wish the authors success.
Reviewer 2 Report
This is a review article summarizing the pharmacological effects and mechanisms of action of resveratrol in the treatment of diabetes and its cardiovascular complications. The review was well-written and balanced.
In Figure 1, the pictures of grapes and red wine may induce misinterpretation, as the pharmacological dose of resveratrol is far greater than the amount from natural products. These pictures are thus better removed from Figure 1.
Reviewer 3 Report
The authors provide a review on the mechanisms of action of resveratrol for the treatment of diabetes and associated CV complications. The in vitro and in vivo pharmacological effects of resveratrol are summarized and the therapeutic potential in the prevention and treatment of diabetes and its CV complications. This is an interesting review article and is very well written and structured Pertinent references have been cited. I have some minor recommendations for consideration.
- The authors need to emphasize and elaborate the novelty aspect of their review article.
- In my opinion, it would be nice if the authors can also provide a table showing clinical studies similar to Table 1, i.e. patient type, dosage, outcome and proposed mechanism. This would strengthen the article and put a better clinical perspective.
- For the preclinical studies presented In Table 1, there is a wide disparity in the dosage used in the studies. Please discuss.
- How does the information presented in this review help to extend/advance current knowledge and assist clinical applicability of resveratrol?
Round 2
Reviewer 1 Report
The revised version of the manuscript is acceptable.
Thank you.
Reviewer 2 Report
The authors responded to the comments properly.
Reviewer 3 Report
The authors have addressed all concerns and have adequately revised their manuscript. I have no further comments.